# Effect of Oral Antidiabetic Drugs on Tuberculosis Risk and Treatment Outcomes: Systematic Review and Meta-Analysis

**DOI:** 10.3390/tropicalmed7110343

**Published:** 2022-10-31

**Authors:** Edinson Dante Meregildo-Rodriguez, Martha Genara Asmat-Rubio, Petterson Zavaleta-Alaya, Gustavo Adolfo Vásquez-Tirado

**Affiliations:** 1Escuela de Medicina, Universidad César Vallejo, Trujillo 13001, Peru; 2Escuela de Posgrado, Universidad Privada Antenor Orrego, Trujillo 13008, Peru; 3Escuela de Medicina, Universidad Privada Antenor Orrego, Trujillo 13008, Peru

**Keywords:** tuberculosis, latent tuberculosis, diabetes mellitus, hypoglycemic agents, metformin, dipeptidyl-Peptidase IV inhibitors, systematic review, meta-analysis, risk factors, risk

## Abstract

Tuberculosis and diabetes mellitus are two global pandemics and rising public health problems. Recent studies suggest that oral antidiabetic drugs (OADs) could reduce the risk of tuberculosis and improve clinical outcomes. However, the evidence is controversial. Therefore, we aimed to assess the effect of OADs on the risk of tuberculosis and treatment outcomes. We systematically searched for six databases from inception to 31 August 2022. We followed a predefined PICO/PECO strategy and included two randomized controlled trials and sixteen observational studies. This study collects 1,109,660 participants, 908,211 diabetic patients, and at least 13,841 tuberculosis cases. Our results show that metformin decreases the risk of active tuberculosis by 40% (RR 0.60; 95% CI 0.47–0.77) in diabetic patients. In addition, metformin exhibits a dose-response gradient (medium doses reduce the risk of active tuberculosis by 45%, while high doses reduce this risk by 52%). On the other hand, DPP IV inhibitors increase the risk of active tuberculosis by 43% (RR 1.43; 95% CI 1.02–2.02). Subgroup analysis showed that study design and metformin dose accounted for the heterogeneity. We conclude that metformin significantly protects against active tuberculosis among diabetic patients. On the contrary, DPP IV inhibitors could increase the risk of developing active tuberculosis.

## 1. Introduction

Tuberculosis (TB) and diabetes mellitus (DM) are two global pandemics of primary concern as public health problems. TB remains an infectious disease that causes significant death in high-risk populations with immunosuppression [1]. Diabetes causes immune dysfunction and increases the risk of TB infection by up to three times [1,2]. Diabetes is prevalent in patients with TB, and poor outcomes of TB control, such as treatment failure, relapse, and even death, are frequent in patients with diabetes [3,4]. 

TB and DM converge to act synergistically, making their control even more challenging. During the recent decades, preclinic and clinical evidence has emerged of the potential beneficial effect of some oral antidiabetic drugs (OADs) on tuberculosis. According to these studies, some OADs—especially metformin—could reduce the risk of latent TB infection (LTBI), active TB, poor treatment outcomes (e.g., mortality), or even poor health-related quality-of-life outcomes in tuberculosis patients with DM [5,6]. However, to date, the evidence is controversial. For example, two recent randomized controlled trials have failed to prove that metformin hastens sputum culture conversion [7] or protects against the development of TB [8].

It is crucial to solve the puzzle of whether OADs have definitive beneficial effects on TB because they could be added to the arsenal of anti-TB drugs, aiding in the goal of TB eradication worldwide. Therefore, we aimed to assess the impact of OADs on the risk of developing tuberculosis diseases and on the treatment outcomes of tuberculosis.

## 2. Materials and Methods

We performed this systematic review following the recommendations of the Cochrane Handbook for Systematic Reviews [9], PRISMA [10], and AMSTAR 2 [11] guidelines. We previously registered the protocol in PROSPERO (CRD42022360949). We searched for observational (cohort, case-control, and cross-sectional) studies and randomized control trials published until 31 August 2022, in Medline (PubMed), Google Scholar, Scopus, ScienceDirect, EMBASE, Web of Science, and Cochrane Library. We combined different keywords, controlled vocabulary terms (e.g., MeSH and Emtree), and free terms according to a PICO/PECO strategy (population: "adults"; exposure: “metformin” OR “sulfonylurea” OR “thiazolidinediones” OR “alpha-glucosidase inhibitors” OR “dipeptidyl-peptidase IV inhibitors” OR “sodium–glucose transporter 2 inhibitors” OR “meglitinide”; comparator: none of the previous oral antidiabetic agents; outcome: “active tuberculosis” OR “latent tuberculosis infection (LTBI)” OR “mortality” OR “sputum culture conversion” OR “tuberculosis recurrence” (Appendix A). We did not limit searches by date or language. 

We included observational studies and randomized control trials and excluded case reports, case series, duplicated publications, and papers in which most patients were <18 years old. Two independent reviewers examined articles, and a third researcher resolved discrepancies. We screened references from retrieved documents for additional articles. 

The articles found were analyzed using the terms of the PICO/PECO strategy and the inclusion and exclusion criteria. In addition, relevant data from each paper were extracted and recorded in a spreadsheet: names of authors, year and country of publication, type of study, number of patients, number of events, measure of association, and adjusted confounders. 

In the meta-analysis, we pooled adjusted odds ratios (ORs), risk ratios (RRs), or hazard ratios (HRs) with 95% confidence intervals (95% CIs) using the generic inverse-variance method. We considered RRs equivalent to the ORs if the frequency of the event of interest was <10% [12]. To convert results of continuous outcomes into dichotomized treatment responses, we used the Cox and Snell method, which allow the direct conversion of standardized mean differences (SMDs) into ORs [13,14]. Forest plots represented the quantitative synthesis. We assessed heterogeneity among studies with Cochran’s Q test and Higgins I^2^ statistic. Heterogeneity was significant (*p*-value < 0.05, I^2^ statistics > 40%), consequently we used a random-effects model. We carried out sensitivity and subgroup analyses. We assessed the risk of bias using the Newcastle–Ottawa scale (NOS) and version 2 of the Cochrane risk-of-bias tool for randomized trials (ROB 2). We examined the publication bias using a funnel plot. 

## 3. Results

We collected 167 studies—136 in the primary search and 31 in the secondary examination. After removing duplicates, there were 90 articles left that we examined in title and abstract. Subsequently, 18 articles remained that were analyzed in full text. We considered these 18 papers for qualitative and quantitative assessment (Figure 1). 

Of the 18 studies included in this review, one was a nested case-control study (CCS), two were randomized controlled trials (RCTs), and sixteen were retrospective cohort studies (RCSs). This review includes 1,109,660 participants, of which 908,211 were diabetic patients and at least 13,841 tuberculosis cases (Table 1).

We only included papers that reported adjusted association measures—OR, RR, HR, or SMDs—and a control group. The lack of adjustment of confounders was the leading cause of the exclusion of most studies (Appendix A). 

Most papers included only diabetic patients. However, we found four articles that included non-diabetic patients; one study only had non-diabetic patients [7], and three others included diabetic and non-diabetic patients [15,16,17]. Therefore, we polled the effect sizes reported only for diabetic patents for almost all the outcomes analyzed. In addition, we examined the effect of metformin on sputum culture conversion separately according to the diabetic status (diabetic patients vs. non-diabetic patients).

**Table 1 tropicalmed-07-00343-t001:** General characteristics of the studies included.

Study	Participants/Cases	Exposition	Outcome	Adjustment of Confounders	OR/RR/HR (95% IC)
Padmapriydarsini C. [7]. 2022. India. RCT.	Non-DM patients ≥ 18 y. Newly diagnosed, culture-positive PTB patients. Standard ATT (control arm) or standard ATT + 1 g daily MET (METRIF arm). Randomized 322. Completed ATT: 155 in METRIF arm and 151 in control arm. Follow-up: 8 weeks.	MET	Time to sputum culture conversion during 8 weeks of ATT.	Age, BMI, gender, smoking, OH, and smear grading.	HR 0.8 (0.624–1.019); *p* = 0.082, for median time to sputum culture conversion after 8 weeks of treatment.
Heo E. [8]. 2021. South Korea. RCS.	N = 76,973 newly diagnosed patients with T2DM, 13,396 MET users, 52,736 MET non-users, and 10,841 excluded patients. TB cases among MET users 46, and among MET non-users 206. Follow-up: 2 y.	MET	Development of TB within 2 y of the index date.	Age, sex, comorbidities, immunosuppressives, CMI, anti-DM treatment, healthcare utilization, hospitalization days, and outpatient visit days.	HR 1.17 (0.75–1.83); *p* = 0.482 for MET and prevention of TB development. HR 1.10 (0.67–1.86) for Q1 of CD of MET. HR 1.69 (1.05–2.71) for Q2 of CD of MET. HR 0.49 (0.20–1.21) for Q3 of CD of MET. HR 0.10 (0.01–0.70) for Q4 of CD of MET. HR 1.19 (0.75–1.87); *p* = 0.46 for SU.
Lee Y.J. [15]. 2018. South Korea. RCS.	N = 499 patients with culture-positive PTB. DM at diagnosis 105, among them 62 were treated with MET. Follow-up: 2 months.	MET	Sputum culture conversion after 2 months of treatment. Recurrence of TB (isolation of MTB, clinical, or radiological evidence).	Sex, statin use, insulin, cancer, AFB smear grade, and drug resistance.	Analysis including only the patients with PTB and DM: OR 2.69 (0.92–7.95); *p* = 0.07 for sputum culture conversion at 2 m. with the use of MET. OR 1.92 (0.42–8.76); *p* = 0.39 for recurrence rate with the use of MET.
Degner N.R. [16]. 2018. Taiwan. RCS.	N = 2416 (DM 699, without DM 1717). DM patients ≥ 13 y with culture-confirmed PTB undergoing treatment 634 (MET 216, non-MET 418). Follow-up: 6 months.	MET within 30 days of starting ATT.	Mortality among DM patients undergoing ATT.	Age, sex, CKD, cancer, cavitary diseases, ATT adherence.	Analysis including only diabetic patients: HR 0.56 (0.39–0.82; *p* = 0.002) for MET and mortality.
Lin H.F. [17]. 2017. Taiwan. RCS.	N = 22,256 adults (≥20 y) newly diagnosed with T2DM and 89,024 persons without DM. TB 3410. (See Footnote **). Follow-up: ≥ 2 y.	MET, SU, TZD, AGI	Risk of TB.	Sex, age, OH, COPD, cirrhosis, OH, hepatitis C, CKD, cancer.	Analysis including only the diabetic cases: HR 0.52 (0.43–0.62) for MET; HR 0.76 (0.63–0.92) for SU; HR 0.75 (0.61–0.93) for TZD; HR 0.55 (0.44–0.67) for AGI.
Lin S.Y. [18]. 2018. Taiwan. RCS	N = 49,028 T2DM patients, MET users (N = 44,002) or MET non-users (N = 5026). Follow-up: 12 y (until death or the end of 2010).	MET	Risk of TB.	DM duration, comorbidities (COPD/CKD), OADs and insulin therapy.	RR 0.24 (0.18–0.32); *p* ≤0.0001 for active TB and MET. RR 1.82 (1.25–2.64); *p* = 0.0016 for active TB and SU. RR 1.36 (1.04–1.79); *p* = 0.0238 for active TB and MEG. RR 1.79 (1.35–2.37); *p* ≤ 0.0001 for active TB and TZD. RR 1.36 (1.05–1.77); *p* = 0.0202 for active TB and AGI.
Lin K.H. [19]. 2020. Taiwan. CCS.	DM patients ≥ 20 years old. 6224 controls and 1556 TB cases. Mean follow-up: 11 y.	Different OADs.	Risk of TB.	Sex, age, urbanization level, length of hospital stay, income, comorbidities.	RR 1.032 (0.887–1.200) for TB in low dose MET users. RR 0.904 (0.732–1.117) for TB in high dose MET users. RR 1.154 (0.995–1.338) for TB in SU users. RR 0.960 (0.809–1.138) for TB in MEG users. RR 0.810 (0.693–0.948) for TB in AGI. RR 0.927 (0.789–1.087) for TB in TZD users.
Lee M.C. [20]. 2018. Taiwan. RCS	Newly diagnosed DM patients. Newly diagnosed DM. A total of 88,866 MET users and 88,866 propensity score-matched MET nonusers. During follow-up, 707 MET users and 807 MET nonusers developed active TB. Total TB cases 1514. Follow-up: ≥ 8 y.	MET	Risk of TB.	Sex, T1DM, age, income, COPD, cirrhosis, cancer, bronchiectasis, etc.	HR 0.84 (0.74–0.96); *p* = 0.013 for active TB among all subjects. HR 0.83 (0.72–0.97) for high-dose MET.
Lee M.C. [21]. 2019. Taiwan. RCS.	N = 5846 diabetic TB close contacts. TB cases among MET users 77, among MET non-users 116, among healthy participants 49. Follow-up: 2 y.	MET	Risk of TB.	Age, male, DM complications, TB history, contact area, local TB incidence, income, etc.	HR 0.73 (0.54–0.98); *p* = 0.035 for risk of incident TB among MET users compared to MET non-users. HR 0.66 (0.49–0.88); *p* = 0.006 for low dose MET. HR 0.59 (014–2.48); *p* = 0.473 for high dose.
Pan S.W. [22]. 2018. Taiwan. RCS	N = 40,179 patients with T2DM, 263 acquired TB over a mean follow-up of 6.1 y. Patients aged <20 y or with a TB diagnosis were excluded (N = 9475). Follow-up: 6.1 y.	MET	Risk of TB.	Age, sex, adapted DCSI, index year, income.	HR 0.337 (0.169–0.673); *p* = 0.002 for MET users compared to SU users. MET < 60 cDDD: reference group. MET 60–219 cDDD: HR 0.860 (0.637–1.161). MET 220–479 cDDD: HR 0.706 (0.485–1.028). MET 480 cDDD: HR 0.319 (0.118–0.863).
Tseng C.H. [23]. 2018. Taiwan. RCS	N = 423,949 newly diagnosed DM patients. Cases (TB) 2336: 360 never-MET users and 1976 MET users. Follow-up: ≥72 months.	MET	Risk of TB.	Age, DM duration, sex, occupation, living region, HT, dyslipidemia, obesity, DM-related complications, OADs, etc.	HR 0.552 (0.493–0.617) for TB and MET users compared with MET never users. HR 1.037 (0.918–1.173) for 1st tercile of CD of MET. HR 0.533 (0.469–0.606) for 2nd tercile of cumulative dose of MET. HR 0.249 (0.215–0.288) for 3rd tercile of CD of MET.
Al-Shaer M.H. [24]. 2018. Qatar. RCS	N = 103 patients with poorly controlled DM and PTB, 72 patients receiving MET. Follow-up: 4 months.	MET	Time to negative smears and the impact of adding MET.	Age, weight, gender, treatment group, ATT dose, AFB load, HbA1c, and total MET daily dose.	OR 0.12 (0.03–0.45) for to sputum smear conversion.
Park S [25]. 2019. South Korea. RCS	T2DM patients aged ≥60 y. N = 12,582 patients among each group (MET vs. SU). TB cases among MET group 79, in SU group 103. Follow-up: 11 y.	MET compared to SU.	Risk of TB.	Age, sex, CCI, comorbidities, medications.	HR 0.74 (0.58–0.95) for TB and MET users compared to SU users. HR 0.63 (0.44–0.91) for males compared to females. MET cDDD <50: HR 0.78 (0.51–1.22). MET cDDD 50–200: HR 0.69 (0.36–1.31). MET cDDD 200–400: HR 0.68 (0.33–1.40). MET cDDD >400: HR 0.20 (0.06–0.61)
Fu C.P. [26]. 2021. Taiwan. RCS.	N = 9750 patients with T2DM. TB cases 47. Follow-up among MET and non-MET users was 2.8–1.8 and 2.6–1.8 y, respectively.	MET	Risk of TB.	Age, sex, FBS, HbA1c, HDL-C, LDL-C, TC, UACR.	HR 0.54 (0.3–0.99); *p* = 0.0475 for TB among MET users compared with non-MET users.
Chen H.H. [27]. 2020. Taiwan. RCS.	Diabetic patients >20 years old. N = 6399 DPP4i users and 6399 DDP4i non-users. Events (TB cases) among DPP4i users 32, among no DPP4i users 24. Mean follow-up 5 y.	DPP4i (not specified which one).	Risk of TB.	Gender, age, DCSI, comorbidities, anti-HT agents, insulin, etc.	HR 1.04 (0.57–1.92); *p* = 0.89 for TB in DPP4i users relative to DDP4i non-users.
FDA [28]. 2010. USA. RCT.	N = 4959 patients taking DPP4i, and 2868 controls. TB among those taking DPP4i 6 and among controls 0. Duration of treatment with DPP4i until the report of TB ranged from 144–929 days.	DPP4i (saxagliptin).	Risk of TB.	Dose	RR * 1.5790 (1.5526–1.6059; *p* < 0.0001 for risk of TB.
Su W.J. [29]. 2018. Taiwan. RCS.	N = 47,740 T2DM patients and matched controls (1:1). Follow-up: 8 y. Cases not reported.	MET vs. SU	Risk of TB.	Age, sex, income, DCSI	RR 0.328 (0.174–0.625) for TB among MET initiators compared to SU initiators.
Bailey CJ [30]. 1212. Phase 3 RCT.	N = 282 T2DM patients. Placebo 68, dapagliflozin (210). TB cases among placebo group 0, TB cases among dapagliflozin group 1. Follow-up: 28 weeks.	Dapagliflozin (1mg, 2.5mg, 5mg)	Risk of TB.	Dose	RR * 1.0313 (0.0425–25.0080); *p* = 0.9849 for risk of TB.

DM: diabetes mellitus, T1DM: type 1 DM, T2DM: type 2 DM, HT: hypertension, TB tuberculosis, PTB: pulmonary TB, LTBI: latent TB infection, MTB: *M. tuberculosis*, OADs: oral antidiabetic drug, MET: metformin, SU: sulfonylurea, DPP4i: DPP4 inhibitor, MEG: meglitinides, AGI: alpha-glucosidase inhibitor, TZD: thiazolidinediones, OH: alcohol consumption, FBS: fasting blood glucose, HbA1c: glycated hemoglobin, HDL-C: high-density lipoprotein cholesterol, LDL-C, low-density lipoprotein cholesterol, TC: total cholesterol, UACR: urine albumin-to-creatinine ratio, CCI: Charlson comorbidity index, BMI: body mass index, CC: cumulative dose, cDDD cumulative defined daily dose, FDC: fixed-dose combination, ST: separate tablets, DCSI score: DM complications severity index score, COPD: chronic obstructive lung disease, CKD: chronic kidney disease, ATT: antituberculous therapy, FBS: fasting blood glucose. *: calculated from data apportioned by the original study. ** Footnote: MET (use 18,936, non-use 3320, use and TB 515, non-use and TB 167), SU (use 18,313, non-use 3943, use and TB 546, non-use and TB 136), TZD (use 4511, non-use 17,745, use and TB 110, non-use and TB 572), AGI (use 5503, non-use 16,753, use and TB 108, non-use and TB 574).

The OADs analyzed were the following: metformin (MET), sulfonylureas (SU), meglitinides (MEG), thiazolidinediones (TZD), alpha-glucosidase inhibitors (AGI), dipeptidyl-peptidase IV inhibitors (DPP4i), and sodium–glucose transporter 2 inhibitors (SGLT2i). 

To perform the dose-response analysis, we considered the reported outcomes according to the definition of “low dose” or “high dose” of the OAD in the primary studies. However, this differed considerably among studies. For example, some studies reported doses in tertiles or quartiles of milligrams of cumulative doses [8,19,23], and others used the definition of “defined daily dose” (DDD) [18,20,21,22,25] recommended by the WHO Collaborating Centre for Drug Statistics Methodology [31]. Following the methodology used for most studies [18,20,21,22,25], we considered three mutually exclusive categories: low doses (<150 DDDs), medium doses (>150 DDDs), and high doses (DDDs > 360).

### 3.1. Metformin

Risk of active TB. Metformin decreases the risk of developing active tuberculosis by 40% (RR 0.60; 95% CI 0.47–0.77) in newly diagnosed and previously diagnosed diabetic patients (Figure 2a). 

The RR for low dose metformin was 0.93; 95% CI 0.80–1.07, for medium dose users was 0.55; 95% CI (0.49, 0.62), and for high dose users was 0.48; 95% CI 0.26–0.87 (Figure 2b). 

A meta-analysis including only those studies with metformin that reported adjustment for statin use [8,16,17,18,19,20,23] showed even greater protection against active tuberculosis (HR 0.57; 95% CI 0.41–0.80) (Figure 2c). 

Sputum culture conversion. Metformin does not affect sputum culture conversion (RR 0.70; 95% IC 0.20–2.47) in diabetics or non-diabetics (Figure 2d).

Risk of LTBI. No good-quality studies addressing this issue were available. We excluded two studies that were included in another meta-analysis [5] because these studies did not adjust for potential confounders. 

Recurrence of tuberculosis. We found only one study [15] evaluating this outcome. Therefore, we could not perform a metanalysis to assess this outcome. 

Mortality. We found only one article [16] evaluating this outcome. So, we could not perform a metanalysis to examine this outcome. 

### 3.2. Other Oral Antidiabetic Drugs (OADs)

Risk of active TB. DPP4i increases the risk of active TB by 43% (RR 1.43; 95% CI 1.02–2.02) (Figure 3a). This metanalysis did not find an association between the risk of tuberculosis and the use of sulfonylureas (RR 1.14; 95% CI 0.82–1.60) (Figure 3b), meglitinides (RR 1.12; 95% CI 0.80–1.58) (Figure 3c), thiazolidinediones (RR 1.06; 95% IC 0.69, 1.63) (Figure 3d), or alpha-glucosidase inhibitors (RR 0.84; 95% IC 0.54–1.31) (Figure 3e). We found only one article [26] evaluating the risk of active TB among users of SGLT 2 inhibitors. Consequently, we could not perform a metanalysis to assess this outcome. 

We did not find any studies evaluating OADs other than metformin regarding the risk of LTBI, sputum culture conversion, recurrence of tuberculosis, or mortality.

All the studies included had a low risk of bias (Table 2). 

The funnel plot suggested publication bias (Figure 4).

## 4. Discussion

According to our results, metformin decreases the risk of developing active TB by 40% (RR 0.60; 95% CI 0.47–0.77)—in newly diagnosed and previously diagnosed and treated—diabetic patients. These findings are consistent with other primary studies [17,18,20,21,22,23,25,26,29] and meta-analyses [5,6]. Nonetheless, at least one study failed to show any benefit of metformin on the risk of active TB [19]. 

Yu X et al. [5] performed a systematic review and meta-analysis aiming to evaluate the impact of metformin prescription on the risk of TB, the risk of LTBI, and treatment outcomes of TB among diabetic patients. They searched three databases and included 6980 tuberculosis cases from 12 observational studies. They found that metformin could decrease the risk of TB among diabetic patients (OR 0.38; 95% CI, 0.21–0.66). However, metformin was not related to a lower risk of LTBI (OR 0.73; 95% CI, 0.30–1.79) in diabetic patients. In addition, metformin during the anti-tuberculosis treatment (ATT) was significantly associated with lower TB mortality (OR 0.47; 95% CI 0.27–0.83) and a higher probability of sputum culture conversion at two months of TB disease (OR 2.72; 95% CI, 1.11–6.69) among patients with diabetes. However, metformin prescription did not statistically reduce the relapse of TB (OR 0.55; 95% CI, 0.04–8.25) in people with diabetes. 

Yu X et al. [5] reported heterogeneity and raised possible explanations for its cause. They also described having performed a sensitivity analysis. However, due to the limited data, they failed to conduct meta-regression, dose-effect assessment, subgroup analysis, or publication bias analysis. They attributed the significant heterogeneity to the different study designs, study populations, and the definition of outcomes. Additionally, this study included four papers that we excluded in our metanalysis due to the lack of adjustment of confounders. 

Zhang M and He JQ [6] conducted a systematic review and meta-analysis to determine the association between metformin use and TB in patients with diabetes. They examined five databases and included 17 observational studies, all of which were at low risk of bias. The pooled analysis showed that metformin use was associated with a significantly lower active TB incidence and mortality among individuals with DM (RR 0.51; 95% CI, 0.38–0.69, *p* ⩽ 0.001) and diabetic patients infected with TB (RR 0.34; 95% CI, 0.20–0.57, *p* ⩽ 0.001), respectively. The authors concluded that metformin use is related to benefits in the prevention and treatment outcomes of tuberculosis among patients with diabetes.

Zhang M and He JQ [6] performed sensitivity and heterogeneity assessments in their metanalysis. They attributed this heterogeneity to unequal DM severity or complications, different dosages or durations of metformin exposure, and disparate smoking habits. They also performed meta-regression and searched for potential publication bias. Nevertheless, they did not carry out subgroup analyses nor examine a possible dose-response effect. Furthermore, this study included nine articles that we excluded in our metanalysis due to the lack of adjustment of confounders.

In our meta-analysis, the heterogeneity was significant (I^2^ > 90%, *p* < 0.05). Subgroup analysis showed that the study design (RCT, RCS, or CCS) and the dose of metformin (low, medium, or high dose) accounted for this heterogeneity (test for subgroups difference: I^2^ > 90%, *p* < 0.10). Because of the small number of studies, we did not perform a meta-regression or subgroup analysis according to other variables. Sensitivity analysis, sequentially excluding studies, did not affect the overall estimate, suggesting good consistency. 

When evaluating the protective effect of metformin against active TB disease in diabetic patients, it is critical to control for multiple covariables or confounders which tend to coexist and could modify the outcome. Diabetic patients have a high prevalence of dyslipidemia and are more likely to receive statins [32]. Besides, three metanalyses have shown that statins independently protect against active TB in patients with and without diabetes [33,34,35]. Consequently, we carried out a meta-analysis including only those studies that controlled for these potential confounders by stratifying statin treatment between metformin users and nonusers, adjusting for confounders during regression analysis, or only enrolling patients who were newly diagnosed with DM [8,18,20,21,22,23,26]. Expectedly, metformin protection against active tuberculosis was even more significant (HR 0.57; 95% CI 0.41–0.80).

The subgroup analysis according to the cDDD showed a dose-response gradient. The higher the metformin dose, the lower the risk of developing active TB. Instead, only medium doses (>150 cDDDs) and high doses (cDDDs > 360) of metformin protect against active TB. Medium doses could reduce the risk of TB by 45%, while high doses could reduce the risk by 52% (Figure 2A). These findings are concordant with several primary studies [8,18,20,21,22,23,25,26]. However, one study failed to show this dose-response effect of metformin on the risk of active TB [19]. To date, no other meta-analyses have examined this dose-response effect of metformin on the risk of active tuberculosis.

According to our meta-analysis, metformin does not affect sputum culture conversion (RR 0.70; 95% IC 0.20–2.47) at two months. Furthermore, the two meta-analyses commented above showed discordant results. The study by Zhang M and He JQ [6] showed similar results to ours (RR 1.08; 95% CI 0.98–1.18, *p* = 0.112). However, our meta-analysis excluded three papers from the study by Zhang M and He JQ [6] due to the lack of confounder adjustment. On the contrary, Yu X et al. [5] reported that metformin significantly promoted sputum culture conversion at two months of TB disease (OR 2.72; 95%CI, 1.11 to 6.69). However, our meta-analysis excluded one paper included by Yu X et al. due to the lack of confounder adjustment. We found no studies that had examined sputum culture conversion with oral antidiabetics other than metformin. We found only one study evaluating the effect of metformin on the recurrence of tuberculosis [15] and mortality [30] in diabetic patients. So, we could not perform a meta-analysis to assess these outcomes. Similarly, we did not find any good-quality studies evaluating metformin and the risk of LTBI. In addition, we excluded two studies included in another meta-analysis [5] because potential confounders were not adjusted. 

A problematic issue of the two meta-analyses discussed above [5,6] is that they combined crude and adjusted effect sizes (OR and RR). Meta-analysis was primarily designed for pooling the effect sizes from RCTs, where the randomization process has controlled for confounders and biases [36]. A meta-analysis of observational (non-randomized) studies has this limitation, which could be overcome by including only adjusted effect sizes [9]. Pooling unadjusted results are more straightforward but not more informative than univariate analysis of the original observational studies [37]. The Cochrane manual recommends using the model estimation that includes the largest number of confounding factors [9] since if unadjusted results are combined, a significant effect could be seen that, when controlled for these covariates, could be reduced or even disappear [38].

Unlike metformin, DDP4i could increase the risk of active TB by 43% (RR 1.43; 95% CI 1.02–2.02). Although clinical evidence is limited, this is pathophysiological plausible, especially for saxagliptin. Like other gliptins used in DM, saxagliptin has immunosuppressive effects. In addition, dipeptidyl dipeptidase IV inhibitors modulate the function of CD26, a similar protein on the surface of lymphocytes. As a result, infections, especially urinary and upper respiratory tract infections, are more frequent in patients taking gliptins than in controls. In 2012, the US Food and Drug Administration (FDA) reported five cases of pulmonary tuberculosis and five cases of pleural tuberculosis attributable to saxagliptin [39,40]. In our meta-analysis of DPP4i, the heterogeneity was moderate (I^2^ = 46%, *p* < 0.0001), but due to the scarcity of studies, it was impossible to perform subgroup, sensitivity, or meta-regression analyses. 

This metanalysis did not find an association between the risk of tuberculosis and the use of sulfonylureas (RR 1.14; 95% CI 0.82–1.60), meglitinides (RR 1.12; 95% CI 0.80, 1.58), thiazolidinediones (RR 1.06; 95% IC 0.69, 1.63), or alpha-glucosidase inhibitors (RR 0.84; 95% IC 0.54–1.31). We found only one article [30] evaluating this outcome among users of SGLT 2 inhibitors. Consequently, we could not perform a metanalysis to assess this outcome. 

We highlight some strengths of our meta-analysis: (1) our search strategy was comprehensive and complete and included more studies than any other systematic review, (2) we only included studies that reported adjusted effect sizes, (3) we only included primary studies that specifically examined clinical outcomes, and (4) we performed sensitivity, subgroup, and dose-response analysis. Therefore, our results are more robust than any other meta-analysis reported before. 

This work has limitations mainly due to the lack of studies: (1) heterogeneity was significant, (2) we were unable to perform subgroup analyses according to other important variables such as age, sex, and the continent of origin of the study because most studies come from Asia, (3) we cannot rule out a possible publication bias against negative studies that did not find a significant association between OADs and outcomes, and (4) we could not perform a meta-analysis for OADs other than metformin and DPP4i.

## 5. Conclusions

In conclusion, our systematic review shows that metformin significantly protects against active tuberculosis among diabetic patients. On the contrary, DPP IV inhibitors could increase the risk of developing active tuberculosis. These findings should be interpreted cautiously because of the study limitations and confirmed in well-designed randomized controlled trials in patients with or without DM. Lower-cost preventative and restorative measures are necessary with the burdens of diabetes and tuberculosis rising globally as pandemics. 

## Figures and Tables

**Figure 1 tropicalmed-07-00343-f001:**
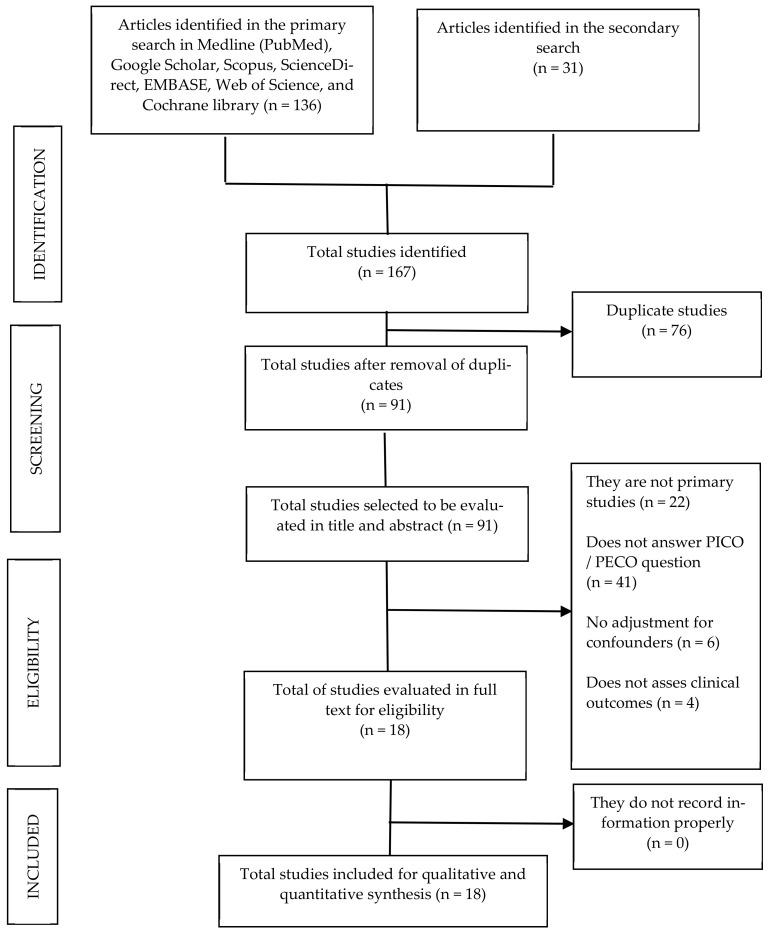
Flow chart of the selection process of the primary studies included.

**Figure 2 tropicalmed-07-00343-f002:**
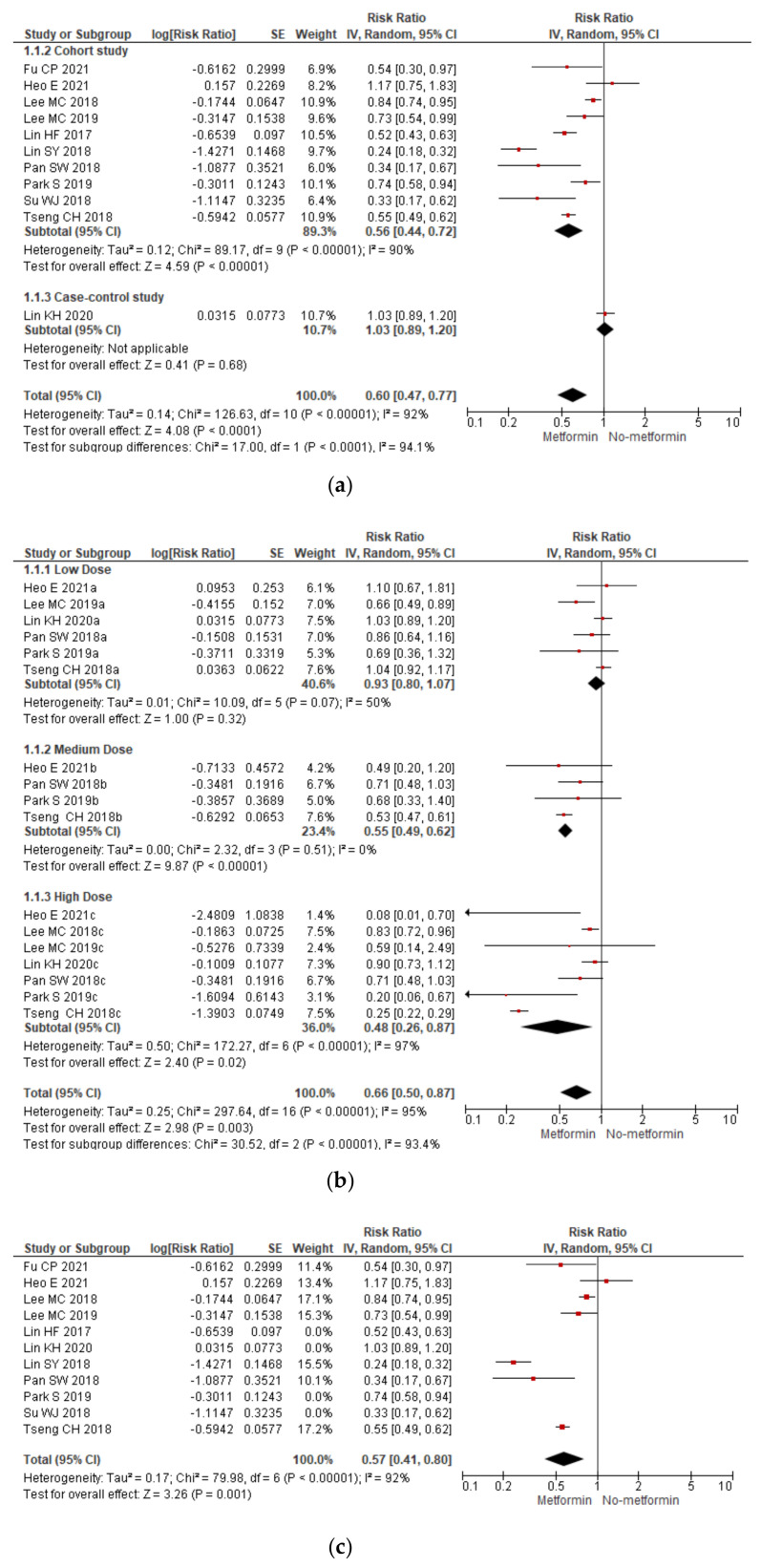
(**a**) Forest plot of the effect of metformin on the risk of active tuberculosis according to the type of study design, (**b**) forest plot of the dose-response effect of metformin on the risk of active tuberculosis, (**c**) forest plot of the effect of metformin on the risk of active tuberculosis including only those studies adjusted for statin use, and (**d**) forest plot of the effect of metformin on sputum culture conversion according to the diabetic status (diabetic patients vs. non-diabetic patients).

**Figure 3 tropicalmed-07-00343-f003:**
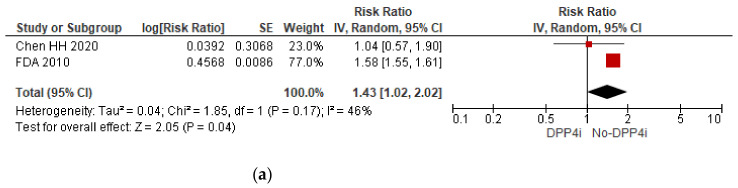
(**a**) Forest plot of the effect of DPP IV inhibitors (DPP4i) on the risk of active tuberculosis, (**b**) forest plot of the effect of sulfonylurea on the risk of active tuberculosis, (**c**) forest plot of the effect of meglitinides on the risk of active tuberculosis, (**d**) forest plot of the effect of thiazolidinediones on the risk of active tuberculosis, and (**e**) forest plot of the effect of alpha-glucosidase inhibitor (AGI) on the risk of active tuberculosis.

**Figure 4 tropicalmed-07-00343-f004:**
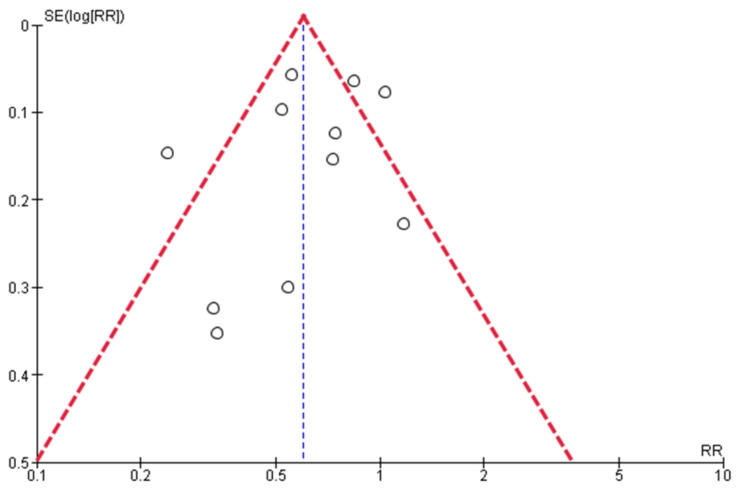
Funnel plot on the effect of metformin on the risk of active tuberculosis.

**Table 2 tropicalmed-07-00343-t002:** Bias assessment of the included primary studies.

Author	Study	Tool	Conclusion
Padmapriydarsini C. [7]. 2022. Multicentric (India) study.	Phase 3 RCT.	ROB 2	Low risk
Bailey C.J. [30]. 1212. Multicentric (USA, Canada, Mexico, Russia, India, South Africa and Puerto Rico) study.	RCT	ROB 2	Low risk
Heo E. [8]. 2021. Korea.	RCS	NOS	Low risk
Lee Y.J. [15]. 2018. South Korea.	RCS	NOS	Low risk
Lin H.F. [17]. 2017. Taiwan.	RCS	NOS	Low risk
Degner N.R. [16]. 2018. Taiwan.	RCS	NOS	Low risk
Lee M.C. [20]. 2018. Taiwan.	RCS	NOS	Low risk
Lin S.Y. [18]. 2018. Taiwan.	RCS	NOS	Low risk
Pan S.W. et al. [22]. 2018. Taiwan.	RCS	NOS	Low risk
Tseng C.H. [23]. 2018. Taiwan.	RCS	NOS	Low risk
Al-Shaer M.H. [24]. 2018. Qatar.	RCS	NOS	Low risk
Park S. [25]. 2019. South Korea.	RCS	NOS	Low risk
Lee M.C. [21]. 2019. Taiwan.	RCS	NOS	Low risk
Fu C.P. [26]. 2021. Taiwan.	RCS	NOS	Low risk
Chen H.H. [27]. 2020. Taiwan.	RCS	NOS	Low risk
FDA [28]. 2010. USA. RCT	RCS	NOS	Low risk
Su W.J. [29]. 2018. Taiwan.	RCS	NOS	Low risk
Lin K.H. [19]. 2020. Taiwan.	CCS	NOS	Low risk

RCT: randomized controlled trial, RCS: retrospective cohort study, CCS: case-control study, ROB 2: version 2 of the Cochrane risk-of-bias tool for randomized trials, NOS: Newcastle-Ottawa Scale (NOS) tool.

## Data Availability

The protocol is available at https://www.crd.york.ac.uk/prospero/display_record.php?RecordID=360949 (accessed on 28 October 2022).

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
