# Peer review of "Effect of Oral Antidiabetic Drugs on Tuberculosis Risk and Treatment Outcomes: Systematic Review and Meta-Analysis"

_tropicalmed, 2022, doi:10.3390/tropicalmed7110343_

Round 1

Reviewer 1 Report

The authors aimed to assess the effect of OHAs on the risk of tuberculosis and treatment outcomes. They selected 18 studies that showed adequate quality among previously published studies on the subject through various literature searches. A meta-analysis was performed using the acquired 999,668 diabetic patients and 11,348 tuberculosis data among them. In conclusion, metformin significantly inhibited the development of tuberculosis in diabetic patients at moderate to high doses, and on the contrary, the DPP4 inhibitor increased the risk of tuberculosis. I understand that the heterogeneity of the studies used in the analysis was high and that it would not be easy to draw clear conclusions as the published studies are concentrated in specific countries. This paper do not bring new knowledge. However, it give more confidence in the tuberculosis protective effect of metformoin.

In addition, although the number of samples is still small to draw conclusions, it also suggested new information that DPP4 inhibitors, one of the diabetes drugs, increase the risk of tuberculosis. Although it does not contribute much to existing scientific knowledge, I think that it is a study that will be helpful for front-line physicians and will be a reference for future research.

Author Response

The authors thank the reviewers for these insightful comments to enrich this paper. This manuscript has been updated according to your valuable comments.

Reviewer 1

The authors aimed to assess the effect of OHAs on the risk of tuberculosis and treatment outcomes. They selected 18 studies that showed adequate quality among previously published studies on the subject through various literature searches. A meta-analysis was performed using the acquired 999,668 diabetic patients and 11,348 tuberculosis data among them. In conclusion, metformin significantly inhibited the development of tuberculosis in diabetic patients at moderate to high doses, and on the contrary, the DPP4 inhibitor increased the risk of tuberculosis. I understand that the heterogeneity of the studies used in the analysis was high and that it would not be easy to draw clear conclusions as the published studies are concentrated in specific countries. This paper does not bring new knowledge. However, it gives more confidence in the tuberculosis protective effect of metformin.

In addition, although the number of samples is still small to draw conclusions, it also suggested new information that DPP4 inhibitors, one of the diabetes drugs, increase the risk of tuberculosis. Although it does not contribute much to existing scientific knowledge, I think that it is a study that will be helpful for front-line physicians and will be a reference for future research.

The authors thank you for this comment to improve our manuscript quality. However, we believe that the scientific content is good, and its main contribution is that it provides more robust evidence than any previous primary study or meta-analysis on this topic.

Reviewer 2 Report

The authors tried to evaluate the Effect of Oral Hypoglycemic Agents on Tuberculosis Risk and Treatment Outcomes: Systematic Review and Meta-analysis. Some similar reviews had published previously and the authors added new published articles with additional evidences. Overall, it is a good article; however, some points need to be addressed carefully.

1.     Oral hypoglycemic agents (OHAs) were less used in the literature recently; instead, the term of oral antidiabetic drugs (OADs) is more adequate.

2.     Some studies enrolled subjects without diabetes mellitus. Subjects with DM and subjects without DM are quite different. Enrolling them (subjects with DM and subjects without DM) into a meta-analysis seems inadequate. It is better to separate the studies with DM and without DM respectively unless you have the reason to mix them together in your meta-analysis.

3.     In conclusion part,”In conclusion, our systematic review shows that metformin significantly protects against active tuberculosis among diabetic patients.” The authors cannot conclude this since some studies enrolled non-DM patients.

4.     In line 37-39, ref 5 and 6 enrolled only DM patients, but the authors described, …… in tuberculosis patients with DM or without DM [5,6]. Please make clarification.

5.     In line 216, line 231, line 240, line 246, cofoundersà confounders

Author Response

The authors thank the reviewers for these insightful comments to enrich this paper. This manuscript has been updated according to your valuable comments.

Reviewer 2

The authors tried to evaluate the Effect of Oral Hypoglycemic Agents on Tuberculosis Risk and Treatment Outcomes: Systematic Review and Meta-analysis. Some similar reviews had published previously and the authors added new published articles with additional evidences. Overall, it is a good article; however, some points need to be addressed carefully.

  1. Oral hypoglycemic agents (OHAs) were less used in the literature recently; instead, the term of oral antidiabetic drugs (OADs) is more adequate.

Thanks for this excellent point, and we agree with you. In this new version, we have made these changes as you suggested.

  1. Some studies enrolled subjects without diabetes mellitus. Subjects with DM and subjects without DM are quite different. Enrolling them (subjects with DM and subjects without DM) into a meta-analysis seems inadequate. It is better to separate the studies with DM and without DM respectively unless you have the reason to mix them together in your meta-analysis.

Thanks for this valuable comment. We did not mix the outcomes of diabetic and non-diabetic patients. However, we must apologize for unintentionally omitting to clarify in Table 1 that, in the three studies that included diabetic and non-diabetic patients, we only included the effect sizes reported for the diabetic group. In this new version, in Results and Table 1 we have added some paragraphs to clarify these issues.

  1. In conclusion part, “In conclusion, our systematic review shows that metformin significantly protects against active tuberculosis among diabetic patients.” The authors cannot conclude this since some studies enrolled non-DM patients.

Having clarified the previous point, we consider that our conclusion is valid.

  1. In line 37-39, ref 5 and 6 enrolled only DM patients, but the authors described, …… in tuberculosis patients with DM or without DM [5,6]. Please make clarification.

We thank you for this comment. You are right; these two metanalyses included only diabetic patients. Consequently, we have made the proper correction in this new version of our manuscript.

  1. In line 216, line 231, line 240, line 246, cofoundersà confounders

We thank you for your valuable comment. We have changed “cofounder” to “confounder”.

Note: Changes in this new version of our manuscript have been highlighted in red.

In addition, we have corrected the total number of participants and tuberculosis cases, including the number of diabetic patients. Since we pooled odds ratios (OR), risk ratios (RR), or hazard ratios (HR) using the generic inverse variance method, not by entering the number of cases and events, our estimates of the global effect did not change.

Checked and corrected

Round 2

Reviewer 2 Report

The authors addressed the points well. Line 38: oral hypoglycemic agents (OADs), should be oral antidiabetic drugs (OADs).